# Understanding the Significance of Biochemistry in the Storage, Handling, Purification, and Sampling of Amphiphilic Mycolactone

**DOI:** 10.3390/toxins11040202

**Published:** 2019-04-04

**Authors:** Jessica Z. Kubicek-Sutherland, Dung M. Vu, Aaron S. Anderson, Timothy C. Sanchez, Paul J. Converse, Ricardo Martí-Arbona, Eric L. Nuermberger, Basil I. Swanson, Harshini Mukundan

**Affiliations:** 1Chemistry Division, Los Alamos National Laboratory, Los Alamos, NM 87545, USA; jzk@lanl.gov (J.Z.K.-S.); dvu@lanl.gov (D.M.V.); aaronsa@lanl.gov (A.S.A.); 2Bioscience Division, Los Alamos National Laboratory, Los Alamos, NM 87545, USA; tim_san@lanl.gov (T.C.S.); rm-a@lanl.gov (R.M.-A.); basil@lanl.gov (B.I.S.); 3Department of Medicine, Johns Hopkins University Center for Tuberculosis Research, Baltimore, MD 21218, USA; pconver1@jhmi.edu (P.J.C.); enuermb@jhmi.edu (E.L.N.)

**Keywords:** mycolactone, storage and handling, amphiphilic, lipoproteins

## Abstract

Mycolactone, the amphiphilic macrolide toxin secreted by *Mycobacterium ulcerans*, plays a significant role in the pathology and manifestations of Buruli ulcer (BU). Consequently, it follows that the toxin is a suitable target for the development of diagnostics and therapeutics for this disease. Yet, several challenges have deterred such development. For one, the lipophilic nature of the toxin makes it difficult to handle and store and contributes to variability associated with laboratory experimentation and purification yields. In this manuscript, we have attempted to incorporate our understanding of the lipophilicity of mycolactone in order to define the optimal methods for the storage, handling, and purification of this toxin. We present a systematic correlation of variability associated with measurement techniques (thin-layer chromatography (TLC), mass spectrometry (MS), and UV-Vis spectrometry), storage conditions, choice of solvents, as well as the impact of each of these on toxin function as assessed by cellular cytotoxicity. We also compared natural mycolactone extracted from bacterial culture with synthesized toxins in laboratory (solvents, buffers) and physiologically relevant (serum) matrices. Our results point to the greater stability of mycolactone in organic, as well as detergent-containing, solvents, regardless of the container material (plastic, glass, or silanized tubes). They also highlight the presence of toxin in samples that may be undetectable by any one technique, suggesting that each detection approach captures different configurations of the molecule with varying specificity and sensitivity. Most importantly, our results demonstrate for the very first time that amphiphilic mycolactone associates with host lipoproteins in serum, and that this association will likely impact our ability to study, diagnose, and treat Buruli ulcers in patients.

## 1. Introduction

Buruli ulcer (BU) infections caused by *Mycobacterium ulcerans* require a rapid diagnosis and initiation of antibiotic therapy at an early stage in order to prevent the devastating consequences of advanced ulceration. The pathogenesis of *M. ulcerans* is dependent on the activity of mycolactone, a plasmid-encoded polyketide-derived macrolide toxin [1,2,3,4]. Mycolactone is responsible for the cytotoxicity [5,6], immunosuppression [7], and painless ulcer formation [8] characteristics of BU disease. The toxin has been detected in the blood and tissue of mice [9,10,11] and patients [12,13,14] prior to ulcer formation, making mycolactone a promising biomarker for BU diagnostics. However, the unique biochemistry of this lipid-like toxin has hampered research in the host-pathogen biology of this disease, and consequently, the development of a simple point-of-care diagnostic tool. There are at least five known structural variants of mycolactone produced by closely-related mycobacteria, all of which contain a highly conserved 12-membered lactone ring and upper side chain, as well as a variable unsaturated fatty acyl lower side chain [1,2,15,16,17,18]. Of these, mycolactone A/B is the most toxic form [19]. Even minor structural alterations to the lower (termed southern side chain) can significantly reduce the biological activity of the toxin [20,21,22,23,24,25], and this includes labeling with dye [20,21] and biotin [26,27] for use in laboratory experiments and studies.

Purification of natural mycolactone in the form of lipid extractions from bacterial culture has been performed by chloroform–methanol (2:1 *v*/*v*) on liquid broth cultures [3] or ethanol on bacterial colonies scraped directly from agar plates [28]. Further purification can then be done using thin-layer chromatography (TLC), whereby the UV-active toxin can be separated from other lipids in the extract [3,29,30]. However, it is known that mycolactone purified from *M. ulcerans* (herein referred to as natural mycolactone) is often contaminated by unknown compounds [17,19]. Total chemical synthesis of mycolactone has provided a means for obtaining robust, homogenous, well-defined samples with high reproducibility [17,21,22,31,32]. However, total synthesis of mycolactone is an extremely complicated process and requires substantial synthetic expertise, thus limiting the global supply of this crucial material [18,19,33].

There is evidence of variability in assay methods, handling conditions, and purification strategies for this lipophilic toxin. Lipids stick to plastic surfaces, aggregate in aqueous solutions, and are sequestered by other lipid-containing molecules [34]. Previous work from our team and others have shown that amphiphilic biomarkers of infection are carried by host lipoproteins in the aqueous milieu of blood, and this sequestration is critical for the transport, toxicity, and pathogenicity of the organism, while influencing immune induction mechanisms as well [34,35,36,37]. It has been suggested that the lipid-like structure of mycolactone associates with other lipophilic molecules, such as lipid membranes [19,38,39], and we hypothesize that lipoproteins constitute a major category of lipophilic host moieties with which mycolactone may interact. We propose that generating antibodies to mycolactone is not only challenging because of its low antigenicity [19], but also owing to the physical interaction of mycolactone with other lipidic molecules in physiological solutions. These interactions must either be accounted for or disrupted in any detection platform in order to improve sensitivity in physiological conditions.

Several investigators have explored the conditions for optimal storage and handling of mycolactone. Marion et al. provided a thorough characterization of mycolactone handling and storage conditions, showing that the toxin is sensitive to photodegradation (50% reduction in toxicity after only 7.5 min of exposure to UV light) and stable at high temperatures (100 °C for 6 h) [40]. Toxin degradation was most profound under UV-irradiation (254–365 nm), and to a lesser extent in sunlight and artificial light. The toxin was found to be stable under red light. Rearrangement in mycolactone chemical structure as a result of photodegradation was shown to significantly reduce cytotoxicity [41]. Storage in glass tubes slowed the photodegradation slightly when compared to quartz tubes, while storage in amber tubes showed partial protection from UV light, and complete protection from photodegradation by natural light [40]. Mycolactone stored in ethanol was also shown to be more stable than the toxin stored in acetone, while storage in aqueous solutions was not examined. Similarly, storage of toxin-containing samples in ethanol proved crucial to the detection of mycolactone in mouse footpads by TLC [10]. Whereas these studies shed some light on optimal conditions for the storage and characterization of mycolactone, a systematic study of the impact of various solvents, storage conditions, and measurement techniques on the detection of mycolactone in laboratory and physiologically relevant matrices does not exist. In this manuscript, we present a systematic assessment of the impact of storage conditions (plastic, glass, and others), solvents (ethanol, water, media, and others), and approach (UV-Vis, mass spectrometry (MS), and TLC) on the measurement of mycolactone A/B (synthetic vs. natural), and expand the findings to assess the interaction of the lipophilic toxin with carrier moieties in host samples. Beyond highlighting the most suitable conditions for laboratory testing of mycolactone A/B, we hope to document the differential association and presentation of the toxin in serum, which is required for the better understanding of host-pathogen biology, as well as the development of diagnostic approaches.

## 2. Results

### 2.1. Purification of Natural Mycolactone Is Inconsistent

Purification of mycolactone in the form of an acetone-soluble lipid extract from *M. ulcerans* bacterial culture is routinely performed in the laboratory [3]. Further purification by TLC is used to separate the UV-active toxin from other lipids found in the bacterial extract [3], followed by characterization using MS [2]. In order to understand the batch-to-batch variability of natural mycolactone samples, we purified 12 independent bacterial extracts by TLC using standard methods [3] and then characterized these extracts by MS, UV-Vis, and cytotoxicity (Table 1, Figure 1). Bacteria were grown on agar plates from which colonies were scraped and then purified using an ethanol extraction method [28]. Four out of twelve samples did not display a UV-active band by TLC (samples 2, 4, 6, and 10). For sample 2, this was thought to be due to a low starting dry weight (0.027 g) of bacteria used in the extraction, resulting in mycolactone levels below the limit of detection by TLC. However, for sample 4, the starting dry weight of bacteria (0.250 g) was similar or greater to that of other samples that yielded detectable concentrations of mycolactone (0.218 g, sample 1). For samples 6 and 10, in the absence of a UV-active band, the area on the TLC plate corresponding with the synthetic mycolactone positive control spot was excised and purified for further analysis. This was not done with samples 2 and 4, therefore there is no corresponding MS or UV-Vis data for these two samples. Interestingly, mycolactone was detected by UV-Vis in all 10 samples tested (including 6 and 10), and by MS in 9 out of 10 samples, with sample 6 being the exception (Table 1, Appendix A). Thus, MS and/or UV-Vis demonstrated presence of the toxin even in extracts where no UV-active band was noticeable by TLC, questioning the use of TLC as a catch-all triage method for identification of the toxin.

In order to determine if the toxin observed by all three methods was biologically active, we assessed the extracts for cytotoxicity in a mammalian cell system. Mycolactone concentration was normalized to 40 ng/mL for all samples as determined by UV absorption (λ_max_ = 362 nm, log ε = 4.29) [29]. We observed high variability in the biological activity of the resulting purified natural mycolactone samples (Figure 1b). Only 4 samples (1, 3, 8, and 9) showed levels of cytotoxicity in L-929 mouse fibroblasts that were similar to synthetic mycolactone A/B. The other 6 samples tested displayed significantly reduced cytotoxicity [29]. Samples 6 and 10, which did not show a UV-active TLC band, did not display cytotoxicity and were not significantly different from the negative control (*P* = 0.42 and *P* = 0.10, respectively) despite the presence of mycolactone in MS (sample 10) and UV-Vis (samples 6 and 10). These results indicate an inherent discrepancy between TLC, MS, and UV-Vis methods for detecting bioactive mycolactone A/B.

One interesting observation was that the MS profiles of the ethanol extracts prior to TLC purification were very similar to synthetic mycolactone A/B (Figure 2a,b). These results indicate the absence of other lipids, at least of those that can be measured under these MS conditions (with a high desolvation temperature of 550 °C), and thereby questions the need for TLC-purification to remove other lipid contaminants in bacterial extracts. In fact, TLC-purification greatly reduced the mycolactone 765 m/z peak to near background levels (acetonitrile blank spectra are included in Appendix A), while increasing the m/z peaks of mycolactone co-metabolites at 429 and 781 [42] (Figure 2c), which was seen in all samples tested (Appendix A). The presence of contaminants is more obvious in samples where the mycolactone concentration is low. Figure 2 demonstrates that the MS profile derived from the ethanol extract of the toxin, without purification by TLC, is similar to that of synthetic mycolactone. However, following TLC purification, the mycolactone concentration is greatly reduced and the other background contaminants and co-metabolites are more visible. These results question the need for TLC purification of natural mycolactone A/B stocks used for research purposes. The bacterial culture and purification process are time-consuming, and TLC appears to greatly reduce the total amount of intact toxin recovered. One reason for this discrepancy is likely the amphiphilic biochemistry of the toxin leading to its association with other lipid-containing molecules, which impacts its recovery from complex biological samples, such as bacterial culture.

### 2.2. Mycolactone Forms Aggregates in Aqueous Solution

Since mycolactone is an amphiphilic molecule (Figure 1a), it has low solubility in aqueous media, causing it to either aggregate, associate with other lipids, or stick to materials like plastic. All of these associations impact measurement of the toxin. In order to understand the behavior of the toxin in aqueous media, we first measured the critical micelle concentration (CMC) of this molecule, which is the concentration at which it forms aggregates in solution [43]. We found that mycolactone A/B forms aggregates in water at room temperature at a concentration of 30–60 µM (Table 2). The addition of a biotin label on the southern chain of the toxin reduces this concentration by 30-fold, thus biotinylated mycolactone A/B will form aggregates at 1–2 µM concentrations. This change in its biochemical properties will alter the way biotinylated mycolactone interacts with other lipid and amphiphilic molecules, including cell membranes, and should be taken into consideration in studies using this modified form of the toxin in research. We speculate that common laboratory methods for labeling and characterizing amphiphiles changes their biochemical behavior, resulting in observations that often lack relevance in physiological conditions.

The CMC of mycolactone will vary based on the temperature and complexity of the media in which it is present. The main goal of this experiment was to highlight that modification to the toxin and the experimental design will impact the behavior of this lipophile, and that can induce variability in our measurements and impact the understanding of its involvement in pathogenesis.

### 2.3. Storage Media, but not Container Material, Affect Mycolactone Measurement by UV-Vis 

We set out to determine the effect of media conditions and storage containers on the ability to measure mycolactone A/B over time. Due to the low yield of mycolactone in a single bacterial extract, synthetic mycolactone samples were used for testing of media condition and container storage conditions. The conditions chosen were those found to be relevant for storage of patient samples including water, ethanol, and culture media [44]. In this experiment, mycolactone A/B was diluted to the same concentration in each of the four test media conditions in a glass vial, and then samples were aliquoted in either plastic polypropylene microfuge tubes, siliconized low-retention microfuge tubes, or other glass vials for comparative evaluation. The samples were stored at room temperature, protected from light, and absorbance measurements at 362 nm were taken over time (from 10 min to 24 h) to determine mycolactone concentration. Interestingly, the storage container made little difference in the absorbance of mycolactone measured at any given time. However, the media conditions had a tremendous impact on mycolactone concentration as measured by UV-Vis (Figure 3).

Samples stored in water showed nearly undetectable levels of mycolactone in as little as 10 min regardless of container material (Figure 3a), suggesting aggregation of the toxin in aqueous matrices. However, samples stored in 100% ethanol showed consistent mycolactone concentration measurements across all time points, up to 24 h, and in all storage container materials tested (Figure 3b). Furthermore, we found that greater than 25% ethanol is required to observe mycolactone concentrations above those seen in PBS alone (Figure 4a). Interestingly, adding sodium dodecyl sulfate (SDS), a known surfactant and amphiphilic molecule, to water at concentrations as low as 1 mM significantly increases the mycolactone measured (Figure 4b). Therefore, aqueous solutions can be used to store mycolactone as long as a surfactant, such as SDS, is present.

Samples stored in cell culture media (EMEM) containing 10% horse serum (as is typical for most media preparations) displayed highly inconsistent results, making mycolactone quantification very difficult (Figure 3c). However, when Eagle’s minimum essential medium (EMEM) was tested without the addition of serum, mycolactone concentrations were similar to those observed in water (Figure 3d). These results indicate that mycolactone interacts with serum components, which affects its ability to be detected by UV-Vis. The time dependence of mycolactone measurements in EMEM containing serum (Figure 3c) indicates a slow interaction of mycolactone with whole serum. This effect normalizes between 5 and 24 h, indicating full sequestration of mycolactone, at which point the measurements are reproducible. This led us to investigate the lipophilic components of human serum with which the toxin interacts, as well as the time dependence of these interactions, as such associations in human serum could also impact host-pathogen biology and pathogenesis of Buruli ulcer.

### 2.4. Mycolactone A/B Binds to Human High- and Low-Density Lipoproteins (HDL and LDL)

Previous work from our laboratory and others has demonstrated a role for human lipoproteins, HDL and LDL, in trafficking lipid-containing molecules in blood [34,35,36,37]. It is known that HDL and LDL play a role in transporting bacterial biomarkers, such as lipopolysaccharide from gram-negative bacteria, lipoteichoic acid from gram-positive bacteria, and lipoarabinomannan from gram-indeterminate bacteria [35,45,46,47]. In fact, HDL is currently being evaluated as a therapeutic for endotoxic shock since it sequesters LPS in blood [46,48]. We wanted to determine whether HDL and LDL also binds mycolactone, which may have significant diagnostic and therapeutic implications.

Normal HDL and LDL concentrations in healthy adults are >0.4 mg/mL and <1 mg/mL respectively [49], so we started our tests using lipoproteins diluted to 0.5 mg/mL. Mycolactone was added to these lipoprotein preparations, and the samples were incubated at room temperature and protected from light with absorbance measurements at 362 nm taken over time as before. As compared to mycolactone in PBS, mycolactone absorbance significantly increased in the presence of HDL in as little as 10 min and remained over the course of 24 h (Figure 5a). We attribute this behavior to the rapid sequestration of mycolactone by HDL. In contrast, mycolactone absorbance only increased significantly after 1 h in the presence of LDL (Figure 5b), indicating a slower sequestration rate of mycolactone by LDL. These results, along with our measurements in EMEM containing whole serum, show differential rates of binding to various lipophilic molecules in serum. When measured in the presence of varying concentrations of HDL and LDL, mycolactone absorbance significantly increased with as little as 0.1 mg/mL HDL (Figure 5c) and 0.25 mg/mL LDL (Figure 5d). These findings demonstrate that lipophilic mycolactone associates with human lipoproteins, and this association may be significant for its biological function, transport, and clearance. To our knowledge this is the first description of mycolactone interacting with human HDL and LDL.

## 3. Discussion

Mycolactone produced by *Mycobacterium ulcerans* is a major diagnostic and therapeutic target for Buruli ulcer. In this study, we characterized the toxin from a biochemical perspective, appreciating its amphiphilicity and speculating on the consequent impact and interactions in aqueous milieu. Expanding on work from other investigators, we defined preferred conditions for the storage, purification, and handling of the toxin, highlighting the discrepancy between various detection methods in their ability to accurately measure mycolactone concentration. CMC measurements indicate that natural mycolactone is likely to behave very differently from modified (e.g., biotinylated) toxin, and that these variabilities should be considered in the design of experiments. Ethanol or a surfactant like SDS can be used to solubilize the toxin in aqueous media in order to enhance its detection by UV-Vis. On the other hand, the storage container material (glass, plastic, or siliconized plastic) proved to have little impact on mycolactone recovery under the conditions tested.

Most significantly, our results demonstrate that lipophilic mycolactone associates with host lipoproteins in human serum and may be present in this sequestered form in physiologically relevant samples. This observation is also supported by the fact that we are unable to measure the toxin in media without serum or lipoproteins. These results help to explain the difficulty in generating antibodies against mycolactone as it is most likely presented in a sequestered form with other lipophilic molecules, including lipoproteins, and other host-derived assemblies [19,38,39]. Amphiphiles are extremely common in biological systems. Most conserved bacterial signatures, those routinely recognized by the host innate immune system (i.e., lipopolysaccharides, lipoteichoic acids, lipoarabinomannans), are amphiphilic in nature. Previous work from our group has shown that these molecules associate with host lipoproteins, which not only impacts their availability for detection but also directly affects the induction of innate immune signaling [34,35,36,37]. Yet, little consideration is often given to the biochemistry and physiological presentation of these moieties when measurements are taken. These associations may be used to develop schemes for the direct detection of mycolactone in serum, using lipoproteins as a means for capturing and concentrating the toxin, a strategy developed by our team termed lipoprotein capture [35]. Further characterization of the interactions between the toxin and host lipoproteins, such as HDL and LDL, will be crucial in order to achieve the level of sensitivity required in a clinical diagnostic tool.

These studies indicate the significance of amphiphilic biomolecules on host-pathogen interactions and associated pathogenesis, and the need for further study of mycolactone specifically. Incorporating the biochemical properties of the molecule in question is critical to an accurate understanding of pathogenesis, and for the development of physiologically relevant diagnostic approaches. In this manuscript, we present new insights into the characterization of amphiphilic mycolactone. Our findings indicate the need for more reliable standards and purification methods, and specifically call for broader efforts to synthesize the toxin for consistency in research and development.

## 4. Materials and Methods 

### 4.1. Materials and Reagents

Synthetic mycolactone A/B was a kind gift of Professor Yoshito Kishi (Harvard University, Cambridge, MA, USA) and biotinylated mycolactone was a kind gift of Dr. Caroline Demangel (Pasteur Institute, Paris, France). Natural mycolactone was obtained from *M. ulcerans* bacterial culture (strains 1615 or 1059). Mycolactone concentrations were quantified by UV-Vis spectrophotometry as described previously (λ_max_ = 362 nm; log ε = 4.29) [29]. Purified human high- and low-density lipoproteins (HDL and LDL, respectively) were purchased from Bio-Rad (catalog no. 5685-2004 and 5685-3204), resuspended in filter-sterilized nanopure water, and stored at 4 °C until use.

### 4.2. Natural Mycolactone Purification

*M. ulcerans* strains 1615 [50] and 1059 [51] were kindly provided by Dr. Pamela Small, University of Tennessee. *M. ulcerans* 1059AL is an autoluminescent derivative of strain 1059 [52]. Bacterial strains were grown on selective Middlebrook 7H11 agar. Colonies from 2 or 3 confluent plates were scraped into a pre-weighed Eppendorf tube and weighed. The material was then put in a foil-wrapped 125 mL glass flask with 10 mL absolute EtOH and stirred for 2 h at room temperature. Using a Pasteur pipette, the dissolved or suspended material was transferred to a pair of glass centrifuge tubes, the tops were crimped, and the tubes were spun at ~3000 rpm for 20 min at 4 °C. The ethanol phase was collected into a 25 or 50 mL round bottom glass flask and dried under nitrogen with heat set between 50–60 °C. The residue was resuspended in 200–300 µL of absolute EtOH and stored at −20 °C. Thin-layer chromatography (TLC) was performed as previously described [2]. Briefly, ethanol extracts from *M. ulcerans* were applied to a silica TLC plate and after drying, mobilized using chloroform:methanol:water (90:10:1 *v*/*v*/*v*) as a solvent system. Mycolactone-containing lipids were extracted as a light-yellow band following visualization under UV light, resuspended in absolute ethanol, and stored in amber glass vials at −20 °C until use.

### 4.3. Electron Spray Ionization Mass Spectrometry (MS)

Ethanol extracts were diluted in acetonitrile and directly perfused into an electrospray ionization source on a Waters Synapt G2S mass spectrometer. The MS conditions were initially optimized to erythromycin, as well as a synthetic mycolactone standard before applying them to the ethanol extracts. The MS conditions were: flow rate 50 µL/min over a 2 min period; capillary voltage 2.98 kV; cone voltage 75 V; desolvation temperature 550 °C; desolvation gas 945 L/h; source temperature 150 °C; acquisition range 100–2000 m/z. Mycolactone A/B was detected by the presence of the more abundant sodium adduct [M + Na]^+^ (m/z 765.5); the protonated molecular ion [M + H]^+^ (m/z 743.5), and the dehydrated protonated molecular ion [M + H − H_2_O]^+^ (m/z 725.5) as described previously [2].

### 4.4. Cytotoxicity Assays

L929 mouse fibroblasts (ATCC CCL-1) were purchased from the American type culture collection and maintained in Eagle’s minimum essential medium (Corning, 10009CV) with 10% horse serum (Sigma-Aldrich, H1270-500ML) in tissue culture flasks and incubated in 5% CO_2_ at 37 °C. In this manuscript, Eagle’s minimum essential medium containing 10% horse serum is referred to as “EMEM”, while media without serum is called “EMEM no serum”. Cells were passaged at least 3 times prior to use in cytotoxicity experiments. 24-well tissue culture plates were seeded with 5 × 10^4^ cells per well and allowed to adhere overnight. Medium was discarded and replaced with 500 µL of fresh EMEM containing the desired concentration of mycolactone. Mycolactone dilutions in EMEM were performed in glass test tubes protected from light. After 48 h of incubation, cytotoxicity was measured using the resazurin-based reagent PrestoBlue™ (A13262, Invitrogen) by adding 50 µL of reagent and incubating at 37 °C for 60 min. Measurements were taken using a SpectraMax Gemini EM plate reader (Molecular Devices, Sunnyvale, CA, USA) with excitation at 560 nm and emission at 590 nm. Relative fluorescence values plotted are the mean ± standard deviation (n = 4 replicate wells).

### 4.5. Mycolactone Quantification by UV-Vis

Synthetic mycolactone A/B stocks at 0.2 mg/mL or 0.1 mg/mL in acetonitrile were diluted 1:20 or 1:10 (respectively) in the desired test media condition in a glass vial and then subdivided into a total of 3 glass vials (National Scientific, C5000-995), 3 plastic tubes (Fisher Scientific, 05-408-137), and 3 siliconized low-retention plastic tubes (Fisher Scientific, 02-681-320) for each test condition (technical triplicates). Samples were then incubated at room temperature protected from light. At various time points ranging from 10 min to 24 h, quantification of mycolactone by UV-Vis absorption at 362 nm was performed using a NanoDrop™ ND-1000 spectrophotometer (Thermo Scientific). Measurements were taken using 2 µL of each sample. Blanks prior to each measurement were performed using the appropriate media conditions in the absence of mycolactone. In the case of HDL and LDL, absorbance of lipoprotein (at the indicated concentration) in the absence of mycolactone was subtracted from the value given for each lipoprotein sample containing mycolactone [Abs_362_ with mycolactone − Abs_362_ no mycolactone]. Experiments were performed at least twice. Values plotted are the mean absorbance values at 362 nm ± standard deviation.

### 4.6. Critical Micelle Concentration (CMC)

The CMC of mycolactone A/B was determined using the detergent critical micelle concentration (CMC) assay kit (ProFoldin, CMC1000) as per the manufacturer’s instructions. Briefly, synthetic (1 mg/mL stock in ethyl acetate) or biotinylated mycolactone (1.125 mg/mL stock in ethanol) was diluted in filter-sterilized nanopure water in 2-fold dilutions ranging from 60–0.03 µM, including a “no mycolactone” negative control in 50 µL volumes in duplicate. Then 50 µL of 1X CMC dye (also diluted in water) was added to each sample, bringing the final mycolactone concentration to 1X at 30–0.015 µM. The samples were incubated at room temperature and protected from light for 30 min. The fluorescence was measured using a SpectraMax Gemini EM plate reader (Molecular Devices, Sunnyvale, CA, USA) with excitation at 360 nm and emission at 465 nm. When in monomeric form, there will be little to no change in fluorescent signal as compared to the negative control. Once the CMC is reached, aggregation will occur, and dye will be incorporated within the micelle resulting in an increased fluorescent emission at 465 nm. Thus, the measured fluorescence values for each mycolactone concentration were divided by the value of the negative control signal to yield normalized fluorescence values (n = 2). In monomeric form, the normalized fluorescent value will equal 1. Therefore, a normalized value greater than 1 indicates micelle formation, which was identified by testing for statistical significance (Student’s *t*-test, *P* < 0.05) pairwise between samples of 2-fold increasing concentrations of mycolactone. The CMC was defined as the 2-fold concentration range in which a statistically significant increase in normalized fluorescence units first appears. Experiments were performed twice.

### 4.7. Statistical Analysis

Absorbance values are plotted as means ± standard deviation. Data were analyzed using Students *t*-test to determine statistical significance between individual groups. A significance level (*P*) of less than 0.05 was considered statistically significant (*** *P* < 0.001, ** *P* < 0.01, or * *P* < 0.05).

## Figures and Tables

**Figure 1 toxins-11-00202-f001:**
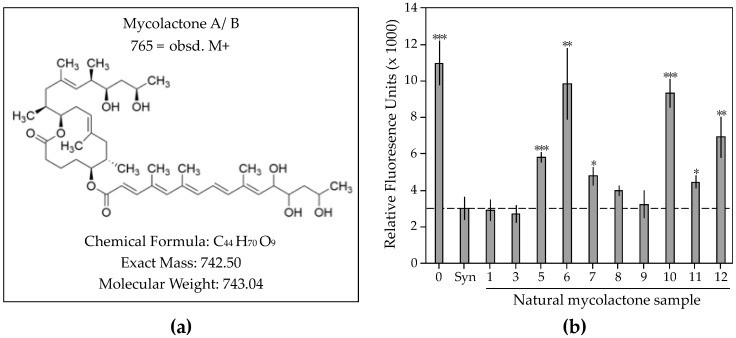
Cytotoxicity comparison of synthetic mycolactone A/B compared to naturally purified mycolactone A/B. The mycolactone toxin (**a**) consists of a highly conserved 12-membered lactone ring, an invariable upper side chain, and an unsaturated fatty acyl lower side chain. (**b**) Cytotoxicity of natural mycolactone purified from *M. ulcerans* was measured using the L-929 mouse fibroblast cell line. Mycolactone concentration was normalized to 40 ng/mL for all samples as determined by UV absorption (λ_max_ = 362 nm, log ε = 4.29). Initial sample concentrations are shown in Table 1. The dotted line indicates cytotoxicity of synthetic mycolactone A/B for comparison. Statistical significance comparing the cytotoxicity of natural mycolactone samples to synthetic mycolactone A/B was determined by Students *t*-test (*** *P* < 0.001, ** *P* < 0.01, * *P* < 0.05).

**Figure 2 toxins-11-00202-f002:**
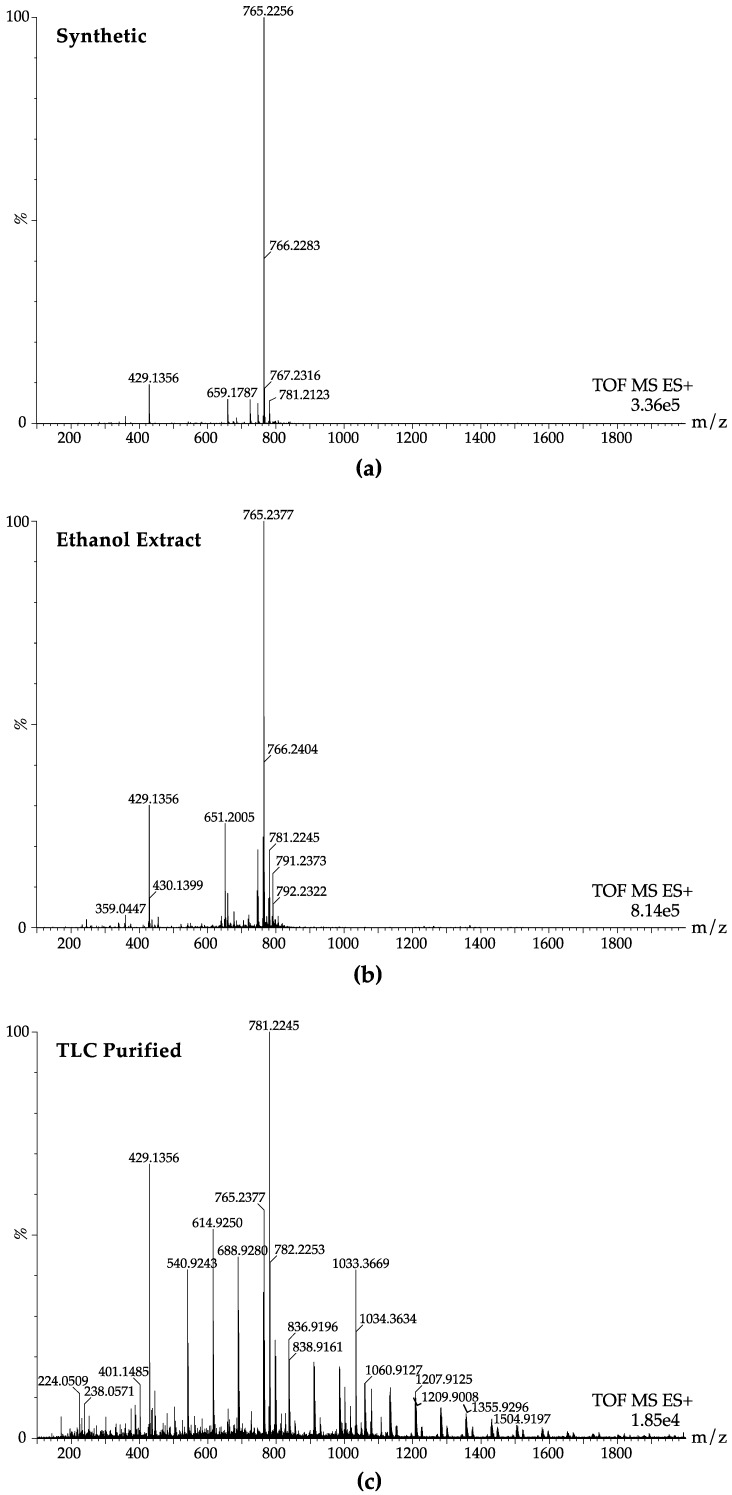
Mass spectrometric comparison of synthetic mycolactone A/B compared to naturally purified mycolactone A/B. There is a characteristic m/z of 765 shown in mass spectrometry profiles of (**a**) synthetic mycolactone A/B as well as for natural mycolactone following (**b**) ethanol extraction and (**c**) TLC purification. For the natural mycolactone sample, four independent ethanol extracts derived from *M. ulcerans* strain 1059AL were pooled together (sample 12, Table 1). Samples were diluted in 100% HLPC-grade acetonitrile and directly perfused into an electrospray ionization source on a Waters Synapt G2S mass spectrometer. Mass spectrometry (MS) profiles for ethanol extracts and TLC purified samples 1–11 are shown in Appendix A.

**Figure 3 toxins-11-00202-f003:**
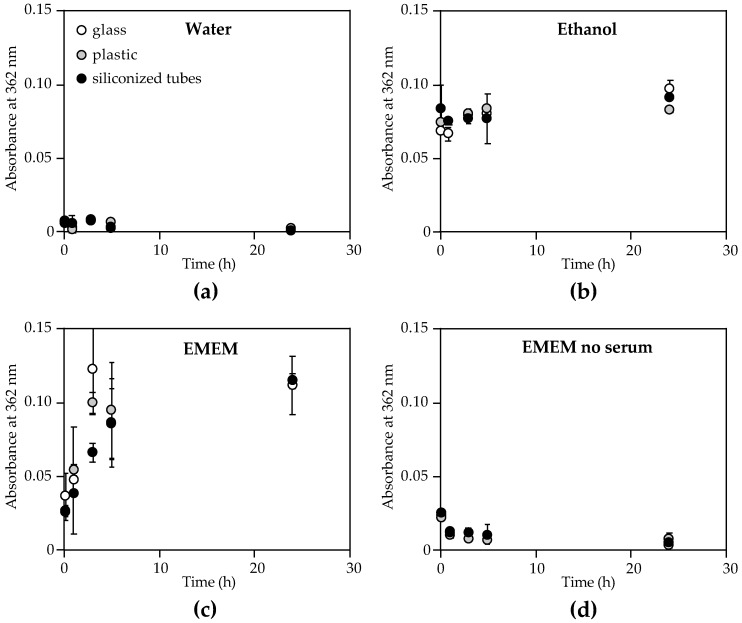
Quantification of mycolactone A/B by UV-Vis absorption after storage in various container materials and media solutions. Synthetic mycolactone A/B (0.2 mg/mL in ethyl acetate) was diluted 1:20 in (**a**) sterile nanopore water, (**b**) absolute ethanol, (**c**) Eagle’s minimum essential medium (EMEM) containing 10% horse serum, or (**d**) EMEM without serum and stored in glass vials (white circles), plastic polypropylene microfuge tubes (grey circles), or siliconized low-retention microfuge tubes (black circles). Samples were stored at room temperature and protected from light until measurements of absorbance at 362 nm were taken at various time points up to 24 h using 2 µL on a NanoDrop™ ND-1000 spectrophotometer (n = 3). Each measurement was blanked using the respective media condition without mycolactone prior to measuring the samples containing mycolactone. Values plotted are mean absorbance at 362 nm ± standard deviation. All experiments were repeated twice.

**Figure 4 toxins-11-00202-f004:**
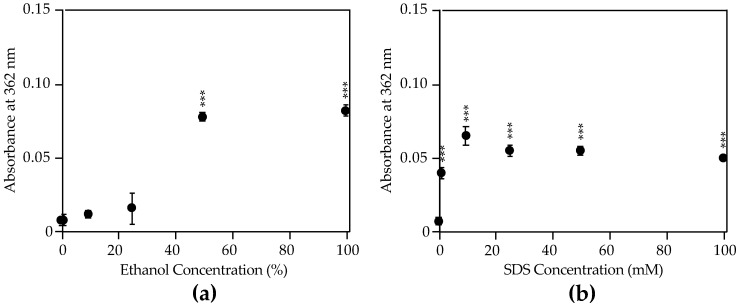
Mycolactone A/B detection by UV-Vis absorption in varying concentrations of ethanol and SDS. Synthetic mycolactone A/B (0.1 mg/mL in ethyl acetate) was diluted in (**a**) ethanol or (**b**) sodium dodecyl sulfate (SDS) at various concentrations in PBS. Samples were stored at room temperature and protected from light for 30 min, at which point 2 µL was used to measure absorbance at 362 nm on a NanoDrop™ ND-1000 spectrophotometer (n = 3). Each measurement was blanked using the same dilution of ethanol or SDS without mycolactone. Siliconized low-retention microfuge tubes were used for sample storage in these experiments. Values plotted are mean absorbance at 362 nm ± standard deviation. All experiments were repeated twice. Statistical significance comparing absorbance of mycolactone in test conditions to PBS was determined by Students *t*-test (*** *P* < 0.001).

**Figure 5 toxins-11-00202-f005:**
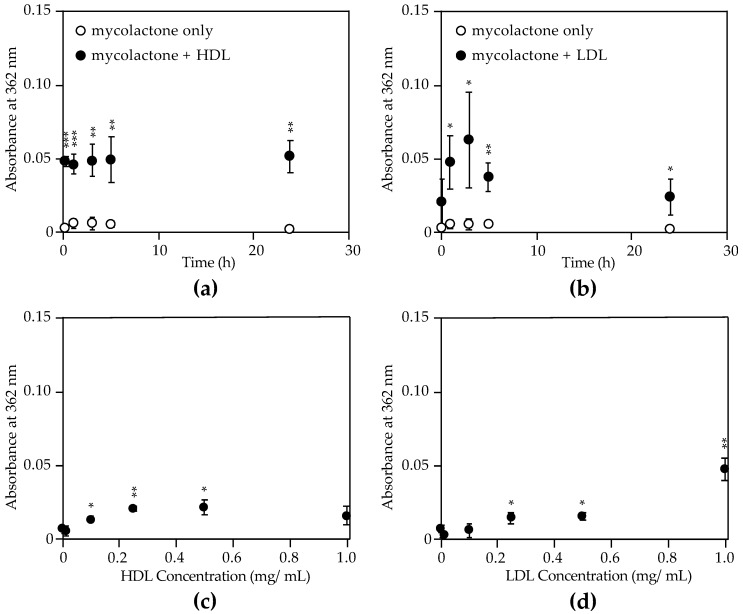
Association of mycolactone with high- and low-density lipoproteins (HDL and LDL). Synthetic mycolactone at a concentration of 0.1 or 0.2 mg/mL in ethyl acetate was diluted 1:10 or 1:20 (respectively) in a PBS solution containing 0.5 mg/mL (**a**) HDL or (**b**) LDL. Samples were stored at room temperature and protected from light for up to 24 h until measurements were taken. Similarly, synthetic mycolactone A/B (0.1 mg/mL) was diluted in PBS containing various concentrations of (**c**) HDL and (**d**) LDL and then stored at room temperature and protected from light for 30 min. Mycolactone was also diluted in PBS in the absence of lipoproteins and stored under the same conditions as a negative control. Siliconized low-retention microfuge tubes were used for sample storage in these experiments. Absorbance at 362 nm was measured using 2 µL on a NanoDrop™ ND-1000 spectrophotometer (n = 3). Measurements were also taken of each lipoprotein (HDL and LDL) at each indicated concentration in the absence of mycolactone, and these absorbance values were subtracted from the value given for each lipoprotein sample containing mycolactone [Abs_362_ with mycolactone–Abs_362_ no mycolactone]. Values plotted are mean absorbance at 362 nm ± standard deviation. All experiments were repeated twice. Statistical significance comparing absorbance of mycolactone with HDL or LDL to mycolactone in PBS alone (no lipoprotein) was determined by Students *t*-test (*** *P* < 0.001, ** *P* < 0.01, * *P* < 0.05).

**Table 1 toxins-11-00202-t001:** Characterization of natural mycolactone A/B from *Mycobacterium ulcerans* lipid extracts.

Sample	*M. ulcerans* Strain	Dry Weight of Bacteria (g)	Volume of Extract (µL)	TLC Band ^a^	MS Peak ^b^	Mycolactone Conc. (µg/mL) ^c^
1	1059AL	0.218	150	+	+	53
2	1615	0.027	25	–	n/a	n/a
3	1059	0.110	100	+	+	17
4	1059AL	0.250	150	–	n/a	n/a
5	1059AL	0.244	235	+	+	8
6	1059AL	0.102	150	–	–	6
7	1059AL	0.153	150	+	+	10
8	1059AL	0.132	150	+	+	19
9	1059AL	0.211	200	+	+	9
10	1615	0.195	260	–	+	9
11	1059	0.130	90	+	+	42
12 ^d^	1059AL	n.d.	375	+	+	11

n.d. not determined; n/a no thin-layer chromatography (TLC) purified product was collected; ^a^ ultraviolet-active component with a refractive index of 0.2–0.4; ^b^ major peak by mass spectrometry at m/z 765; ^c^ measured by absorbance at λ_max_ = 362 nm (log ε = 4.29); ^d^ Sample 12 contains four pooled ethanol extracts from *M. ulcerans* strain 1059AL.

**Table 2 toxins-11-00202-t002:** Critical micelle concentration (CMC) of mycolactone A/B in water.

Mycolactone	CMC ^a^	Fold-Change ^b^
Synthetic	30–60 µM	−
Biotinylated	1–2 µM	0.03

^a^ determined in filter-sterilized nanopore water at room temperature. ^b^ relative to synthetic mycolactone A/B.

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
