# Peer review of "Understanding the Significance of Biochemistry in the Storage, Handling, Purification, and Sampling of Amphiphilic Mycolactone"

_toxins, 2019, doi:10.3390/toxins11040202_

Round 1

Reviewer 1 Report

This manuscript described the measurement of mycolactonede via the variability of TLC, MS and UV-Vis. The methods are adequately described and the results are clearly presented. Also this toxin was associated with host lipoproteins in serum for the first time, which could potentially impact the development of diagnostic and treatment strategies for Buruli ulcer. 

Author Response

Reviewer 1 did not have any comments for us to address.

Reviewer 2 Report

- The first section in results concerning the purification of mycolactone from M. ulcerans had weaknesses and  conclusions cannot be drawn from these experiments.

It is inconsistent to compare the mycolactone extraction of 12 bacterial extracts from 3 different strains with different weights of bacteria and a different volume of extract. It is important to start with a same culture batch of M. ulcerans and prepare aliquot of bacteria to have similar extracts and compare several conditions.

The referent model of extraction of mycolactone from M. ulcerans culture is a chloroform/methanol (2/1) extraction following a folch reaction and a precipitation of phospholipids in cold acetone. Authors need to demonstrate whether this method is consistent for mycolactone extraction. They need to compare this method with their method of extraction using Ethanol.

Authors conclude that TLC is not necessary to purify mycolactone from bacteria while they have not sufficient proof of evidence of the purity of mycolactone after ethanol extraction. One of an important point is the use of a very weak quantity of bacteria to extract mycolactone, hiding the presence of contaminant in ethanol extract before TLC. Furthermore, they need to perform at least RMN analysis to show the purity of mycolactone extracts.

For the evaluation of the cytotoxicity, authors need to precise the quantity of mycolactone used on the cells for each condition.

- Next sections of the results

It is very usefull to have determined the CMC of mycolactone in water. Is it possible to try to determine the CMC in other aqueous buffers and in solvents like ethanol?

Authors demonstrated that mycolactone associates with liproproteins like HDL or LDL in aqueous buffer. It could be interesting to show if mycolactone associated with these lipoproteins could be solubilized in solvent or not.

Author Response

Dear Reviewer

Thank you for your insightful comments and suggestions. We have attempted to address them to the best of our ability. The detailed responses are in the attached file. Please let us know if you have further comments. We thank you for your review.

Best wishes

harshi (corresponding author)

Round 2

Reviewer 2 Report

The clarification of the choice of method of extraction of mycolactone is based on personal communication and it is difficult to base all experiments on this information. Mycolactone can be easily extracted with chloroform/methanol method from M. ulcerans growing on agar plates as it is already published.

The problem of the low rate of mycolactone extraction is due to the very low quantity of bacteria (dry weight < 0.3g) obtained in culture and could be resolved by culturing M. ulcerans in larger agar plate.